# Multifactor Citation Analysis over Five Years: A Case Study of SIGMETRICS Papers

Eitan Frachtenberg

Department of Computer Science, Reed College, 3203 Woodstock Blvd, Portland, OR 97202, USA; eitan@reed.edu

**Abstract:** Performance evaluation is a broad discipline within computer science, combining deep technical work in experimentation, simulation, and modeling. The field's subjects encompass all aspects of computer systems, including computer architecture, networking, energy efficiency, and machine learning. This wide methodological and topical focus can make it difficult to discern what attracts the community's attention and how this attention evolves over time. As a first attempt to quantify and qualify this attention, using the proxy metric of paper citations, this study looks at the premier conference in the field, SIGMETRICS. We analyze citation frequencies at monthly intervals over a five-year period and examine possible associations with myriad other factors, such as time since publication, comparable conferences, peer review, self-citations, author demographics, and textual properties of the papers. We found that in several ways, SIGMETRICS is distinctive not only in its scope, but also in its citation phenomena: papers generally exhibit a strongly linear rate of citation growth over time, few if any uncited papers, a large gamut of topics of interest, and a possible disconnect between peer-review outcomes and eventual citations. The two most-cited papers in the dataset also exhibit larger author teams, higher than typical self-citations, and distinctive citation growth curves. These two papers, sharing some coauthors and a research focus, could either signal the area where SIGMETRICS had the most research impact, or they could represent outliers; their omission from the analysis reduces some of the otherwise distinctive observed metrics to nonsignificant levels.

**Keywords:** SIGMETRICS; bibliometrics; factors in citation

## 1. Introduction

SIGMETRICS is the flagship conference of the eponymous Association for Computing Machinery's (ACM) special-interest group (SIG) on performance evaluation [1]. First convened in 1973, SIGMETRICS is one of the longest-running conferences in computer science (CS) and has amassed some 2000 published papers, each cited on average at least 28 times [2]. Its scope centers on the relatively focused area of computer performance evaluation, but at the same time, its methods and techniques span a diverse field of analysis, simulation, experiment design, measurement, and observation. This duality is reflected in the content of SIGMETRICS's papers, spanning the gamut from mathematical proofs to practical measurement aspects and all scales from embedded processors to the largest compute clouds. Given the distinctive scope of the conference, as well as its long history, respectable citation rate, and diversity of methods, we may ask: what factors affect the citation count of SIGMETRICS papers?

For better or worse, citations occupy a central role in the bibliometric evaluation of journals, conferences, institutes, and individual researchers [3]. This study is not concerned with the merits and deficiencies of citation analysis. Instead, the aim of this study is to understand the specific citation patterns of this distinctive conference, and compare it to similar conferences. This paper starts with the assumption that citations are a widely used metric of scholarly impact and investigates variations in citations in the context of SIGMETRICS. Specifically, we address the following research questions:

- RQ1: What is the distribution of paper citations after five years?
- RQ2: How have citations evolved over this period?
- RQ3: How many citations are self-citations?
- RQ4: What SIGMETRICS keywords are associated with particularly high citations?
- RQ5: What other factors are related to citations?

The bibliometrics literature is rich with studies looking at these and similar questions in various other disciplines and fields, even within CS [4–7]. To the best of our knowledge, no prior study has looked at the field of performance evaluation, and in particular, the SIGMETRICS conference with its unique characteristics.

As a case study in the field, we started measuring citations of SIGMETRICS and related conferences in a single publication year, 2017. This focus permitted both the collection of fine-grained citation data at monthly intervals and a retrospective look at citations five years since publication. This duration is long enough to allow papers to be discovered, read, cited, and even expanded upon by other scientists, and has therefore been used in a number of related studies [7–9].

This singular focus also permitted the labor-intensive manual collection of multiple associated conference and author factors, such as author demographics and research experience, cleanup of the papers' full text, and counting of self-citations. In the next section (Section 2), we describe in detail our data collection methodology, including the manual assignment of genders to authors to avoid the well-known issues of name-based gender inference. In the results section (Section 3), we enumerate our findings, organized by research question, and then summarize an answer to each of the questions in the context of the previous work related to each question. Finally, we discuss our results (Section 4) and offer some conclusions and directions for future research (Section 5).

## 2. Materials and Methods

To answer these research questions, we collected citation data at regular intervals from a set of 2017 conferences, on which we performed various statistical analyses. Our main dataset is the complete collection and full text of accepted research papers from SIGMETRICS'17. That year, the conference published 27 papers out of 203 submissions (13.3% acceptance rate). Although the final conference proceedings were not published as open access, these papers were freely available during the week of the conference, and authors were permitted to post versions on personal web sites and preprint archives. All papers are still accessible as free e-prints via Google Scholar (GS), and this availability itself had been sometimes linked to higher citation counts [10,11].

Since SIGMETRICS'17 is not covered by the Scopus database, we collected all citation metrics from GS; every month, we recorded the number of citations of each paper, as well as the availability of an e-print. GS is an extensive database that contains not only peer-reviewed papers, but also preprints, patents, technical reports, and other sources of unverified quality [12]. Its citation metrics therefore tend to be higher than those of databases such as Scopus and Web of Science, but not necessarily inferior for paper-to-paper comparisons [13,14]. As we are primarily interested in relative citation metrics, even if the GS metrics appear inflated compared to other databases, we should still be able to examine the relationship between relative citation counts and various other factors. And of course, the free availability of GS data makes our dataset (and that of comparable studies) easier to obtain, verify, and reproduce.

In addition to paper data, we collected basic demographic data for all 104 SIGMETRICS'17 authors (101 unique). Conferences do not generally share (or even collect) demographic data on all authors, so we relied instead on a manual Web search of every author. From authors' email addresses and using regular expressions, we can roughly categorize each author as either affiliated with an education institution (71), industry (12), government (2), or unknown (19). We can also guess their country of affiliation, with nearly half of the authors (50) from North America, some from Europe (15), some from East Asia (6), and most of the rest unknown (31).

Another interesting demographic to observe is *perceived gender* at time of publication [15]. Gender is a complex, multifaceted identity [16], but most bibliometric studies still rely on binary genders—either collected by the journal or inferred from forename—because that is the only designator available to them [15,17–23]. In the absence of self-identified gender information for our authors, we also necessarily compromised on using binary gender designations. We therefore use the gender terms "women" and "men" interchangeably with the sex terms "female" and "male". Using web lookup, we assigned all authors a gender whenever we found a recognizable gendered pronoun or absent that, a photo. This labor-intensive approach was chosen because it can overcome the limitations of automated gender-inference services, which tend to be less accurate for non-Western names and women [18,24,25].

Finally, we also collected proxy metrics for author research experience. Conferences also do not generally offer this information, but we were able to unambiguously link 74 of the authors in our dataset to a GS author profile, from which we recorded their total prior publications and h-index near the time that SIGMETRICS'17 took place.

*Statistics*

For statistical testing, group means were compared pairwise using Welch's two-sample *t*-test and group medians using the Wilcoxon signed-rank test; differences between distributions of two categorical variables were tested with the $\chi^2$ test; and correlations between two numerical variables were evaluated with Pearson's product-moment correlation coefficient. The relative effect of different factors on citations has been evaluated using linear regression. All statistical tests are reported with their *p*-values. All computations were performed using the R programming language with the Quanteda and other packages, which can be found in the source code accompanying this paper.

## 3. Results

This section explores our research questions in detail while bringing in the context of previous research and findings. Following the empirical results for the five research questions, we summarize and aggregate the various factors by using a linear regression model of paper citations.

### 3.1. RQ1 What Is the Distribution of Citations after Five Years

Table 1 shows all 27 papers from SIGMETRICS'17 and their total citations exactly five years since publications, averaging 38.3 citations per paper (median 32). The distribution of citations is also shown as a log-scale density plot in Figure 1.

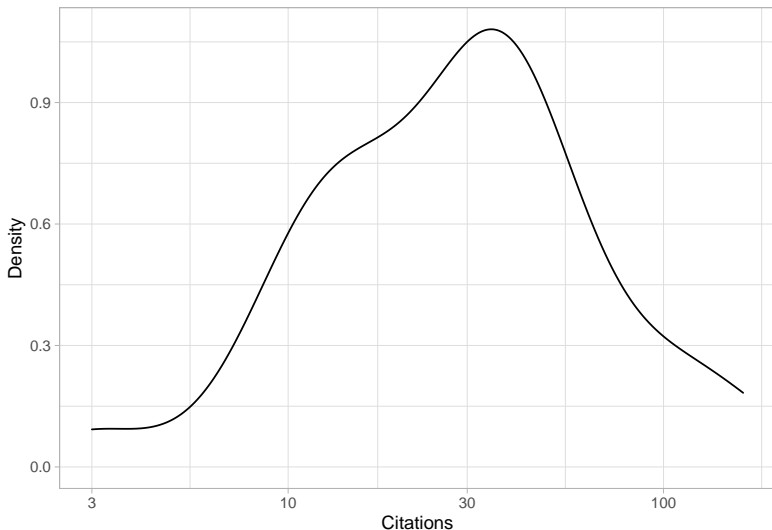

**Figure 1.** Density plot for five-year citation counts distribution, logarithmic scale.

**Table 1.** All SIGMETRICS'17 research papers, ordered by citations after exactly five years.

| # | Citations | Authors | Paper Title |
|---|---|---|---|
| 1 | 163 | 10 | Understanding Reduced-Voltage Operation in Modern DRAM Devices: Experimental Characterization, Analysis, and Mechanisms |
| 2 | 112 | 8 | Design-Induced Latency Variation in Modern DRAM Chips: Characterization, Analysis, and Latency Reduction Mechanisms |
| 3 | 95 | 3 | Dandelion: Redesigning the Bitcoin Network for Anonymity |
| 4 | 63 | 3 | Portfolio-driven Resource Management for Transient Cloud Servers |
| 5 | 52 | 3 | Quality and Cost of Deterministic Network Calculus - Design and Evaluation of an Accurate and Fast Analysis |
| 6 | 48 | 1 | Expected Values Estimated via Mean-Field Approximation are 1/N-Accurate |
| 7 | 44 | 4 | Optimal Service Elasticity in Large-Scale Distributed Systems |
| 8 | 41 | 4 | Using Burstable Instances in the Public Cloud: Why, When and How? |
| 9 | 39 | 3 | Optimal Posted Prices for Online Cloud Resource Allocation |
| 10 | 39 | 5 | Deconstructing the Energy Consumption of the Mobile Page Load |
| 11 | 35 | 3 | A Low-Complexity Approach to Distributed Cooperative Caching with Geographic Constraints |
| 12 | 35 | 1 | Stein's Method for Mean Field Approximations in Light and Heavy Traffic Regimes |
| 13 | 32 | 4 | Overcommitment in Cloud Services: Bin Packing with Chance Constraints |
| 14 | 32 | 5 | Characterizing and Modeling Patching Practices of Industrial Control Systems |
| 15 | 30 | 4 | Outward Influence and Cascade Size Estimation in Billion-scale Networks |
| 16 | 28 | 6 | A Simple Yet Effective Balanced Edge Partition Model for Parallel Computing |
| 17 | 21 | 4 | Persistent Spread Measurement for Big Network Data Based on Register Intersection |
| 18 | 19 | 3 | Investigation of the 2016 Linux TCP Stack Vulnerability at Scale |
| 19 | 19 | 3 | A Case Study in Power Substation Network Dynamics |
| 20 | 15 | 2 | Security Game with Non-additive Utilities and Multiple Attacker Resources |
| 21 | 13 | 4 | Hadoop on Named Data Networking: Experience and Results |
| 22 | 13 | 3 | On Optimal Two-Sided Pricing of Congested Networks |
| 23 | 12 | 4 | Exploiting Data Longevity for Enhancing the Lifetime of Flash-based Storage Class Memory |
| 24 | 12 | 1 | Accelerating Performance Inference over Closed Systems by Asymptotic Methods |
| 25 | 10 | 5 | Queue-Proportional Sampling: A Better Approach to Crossbar Scheduling for Input-Queued Switches |
| 26 | 8 | 5 | Analysis of a Stochastic Model of Replication in Large Distributed Storage Systems: A Mean-Field Approach |
| 27 | 3 | 3 | Hieroglyph: Locally-Sufficient Graph Processing via Compute-Sync-Merge |

Total citations exhibit a typical long-tailed distribution [7,26–28], with two top papers (ostensibly from related research groups) picking up 26.6% of the total citations. However, the adage that "most papers aren't cited at all" [29,30] does not appear to hold for this conference. Moreover, if we compare to five-year citations of papers in natural sciences and engineering only, where about a quarter of the papers remain uncited [9], SIGMETRICS'17 fared much better with no uncited papers—even when omitting self-citations. Although uncited papers are not are as rare as they used to be [9,28], they are starkly absent from SIGMETRICS'17.

We can compare SIGMETRICS's citations to some of its contemporaneous peer conferences.[1] Table 2 shows the mean and median number of citations for two other performance-evaluation conferences from the same ACM special-interest group, IMC and ICPE, as

well as three other well-cited flagship conferences for other ACM SIGs: SIGCOMM [5,6], SIGIR [4], and SIGMOD [7]. Within the relatively narrow field of performance evaluation, SIGMETRICS sits somewhere between ICPE and IMC in terms of mean citations. However, in the flagship conferences of the much larger subfields of communications, information retrieval, and management of data, mean citations are much higher than in SIGMETRICS and display even longer tails. Indeed, the most-cited papers in these three conferences garnered 741, 799, and 647 54-month citations respectively, compared with SIGMETRICS's 142. Even their standard deviation is much higher, despite their larger sample sizes (number of papers).

**Table 2.** Mean and median citations after 54 months for SIGMETRICS and other contemporaneous conferences.

| Conference | Papers | Mean | Median | Std Dev |
| --- | --- | --- | --- | --- |
| SIGMETRICS | 27 | 34.2 | 26.0 | 32.0 |
| ICPE | 29 | 16.5 | 13.0 | 15.8 |
| IMC | 28 | 49.9 | 33.5 | 37.2 |
| SIGCOMM | 36 | 135.2 | 104.5 | 146.3 |
| SIGIR | 78 | 73.7 | 28.0 | 131.9 |
| SIGMOD | 96 | 49.4 | 30.5 | 76.3 |

*3.2. RQ2: How Have Citations Evolved over Time?*

Observing the total citations of papers at a given time point offers only a static view of a metric that is inherently a moving target. Citations tend to follow different dynamics, often accelerating first as papers are discovered, and then decelerating as their novelty recedes and as different papers, disciplines, and fields, exhibit very different ageing curves [31,32].

After five years, all SIGMETRICS'17 papers likely had a chance to be discovered by fellow researchers, as evidenced by the fact all are cited by outside researchers. We can therefore ask questions such as: how are citations changing over time? what is the citation velocity of different papers? have any papers already peaked after five years and show a decrease in citation velocity?

As Figure 2 shows, different papers do indeed accumulate citations at different rates, ranging from about zero to three additional citations per month. Some papers even show temporary dips in citations, variations in counting which are not unusual for GS [12]. There is even a months-long gap for paper #3 when this paper could not be found at all on GS.

An interesting observation is that citation velocity appears fairly constant for many papers, contradicting our expectation of accelerated growth. This observation can be noticed more readily when looking at the month-to-month difference in citation counts (Figure 3). Papers #1 and #2 (the top-cited papers) both show rapid growth through the first fifteen months or so, and then diverge, with paper #2 showing a slow decline in citation growth. Papers #3 and #4 exhibit somewhat erratic growth over time, likely because of the temporary artifacts we observed in their GS data. Most other papers show a fairly stable growth rate hovering around one new citation per month or two. None of the papers appear to have clearly peaked yet, at least in the period examined.

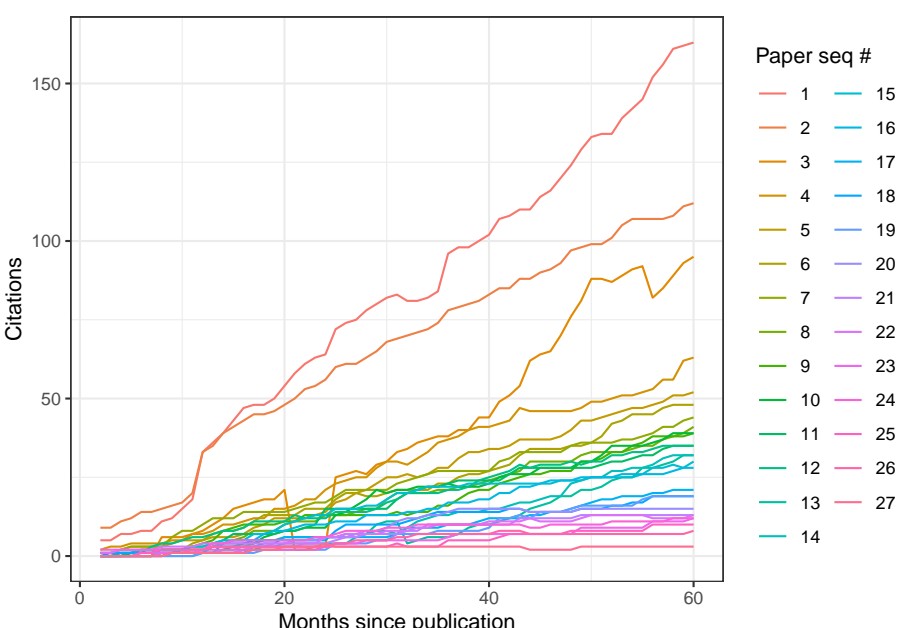

**Figure 2.** Total citations over five years, sampled monthly.

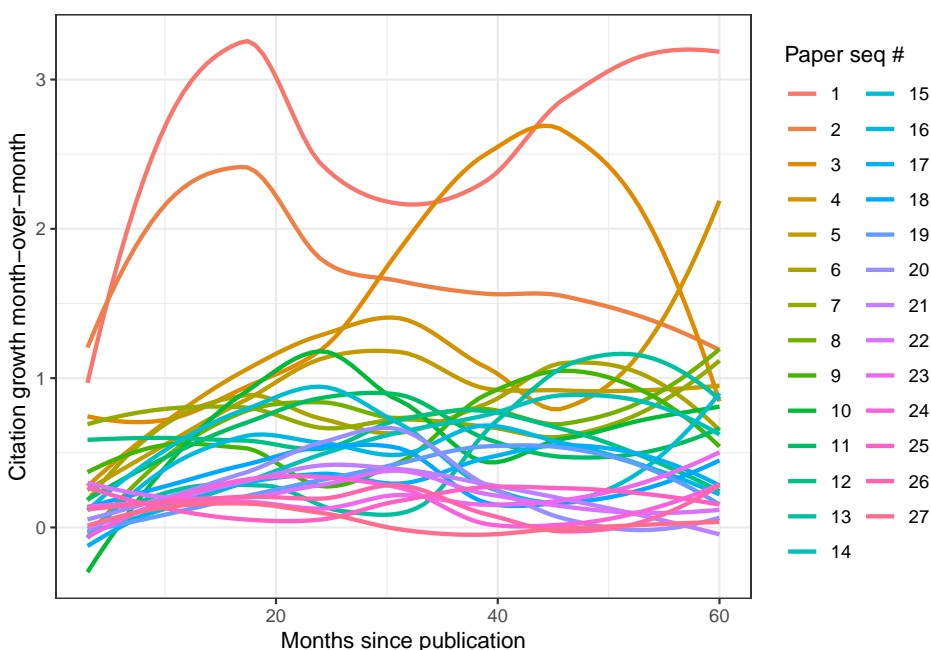

**Figure 3.** Monthly citation growth over five years using LOESS smoothing.

As another verification of this constant growth rate, we modeled a simple linear regression to each paper using citations as the outcome variable and months-since-publication as the only predictor variable. All but the last two (least cited) papers measured an adjusted $R^2$ value above 0.9, averaging 0.94 overall. This near-constant growth appears to be typical for the field. Looking at the other two sibling conferences (Figure 4) shows a similar picture of linear growth. On the other hand, a few SIGMOD papers show accelerating citation growth, more in-line with our expectations from past results. It is possible that the smaller field of performance evaluation offers little opportunities for exponential growth, since results are likely disseminated to the entire research community at about the same time.

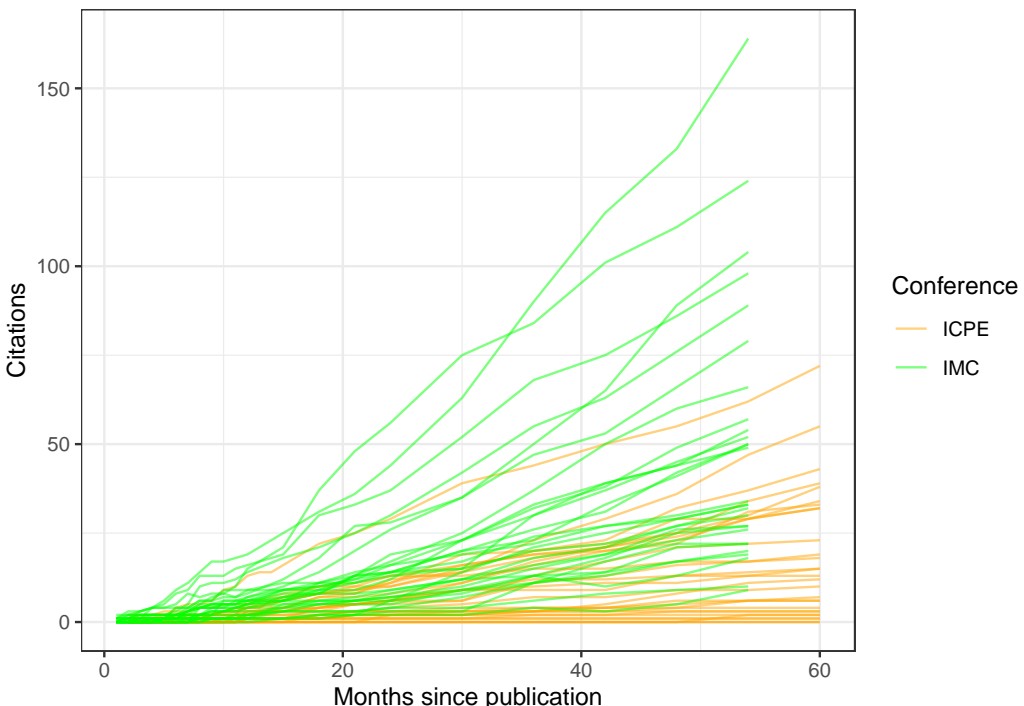

**Figure 4.** Total citation growth for ICPE and IMC, sampled biannually.

### 3.3. RQ3: How Many Citations Are Self-Citations?

Self-citations are fairly common in the sciences, and have been estimated to comprise 10–40% of all scientific production, depending on field [33–35]. On the one hand, self-citations represent a natural evolution of a research team's work, building upon their previous results, especially in systems projects that often involve incremental efforts of implementation, measurement, and analysis [35]. On the other hand, self-citations can be problematic as a bibliometric measure of a work's impact, because they obscure the external reception of the work and are prone to manipulation [36].

The amount of self-citations in SIGMETRICS'17 (Figure 5) varies from 0 to 88, averaging 11.48 per paper (24.25% of all citations; SD: 22.45), agreeing perfectly with the 24% rate found for CS papers in Norway [33]. The same study also found a high ratio of self-citing papers overall, agreeing with our data where all but 3 papers include at least one self-citation.

At first blush, SIGMETRICS'17 self-citations appear to be strongly correlated with total citations (Pearson's $r = 0.83$, $p < 10^{-7}$), suggesting that self-citations represent a meaningful fraction of total citations [35]. However, the high variance in this ratio across papers (right column) contradicts this hypothesis. A more likely explanation for this high correlation is that it is skewed by the heavy-tail papers. Omitting the top two papers alone weakens the correlation to nonsignificant levels ($r = 0.37$, $p = 0.07$).

The two most-cited papers are also the two with the largest author teams, posing the question of whether this factor can better explain self-citations [34]. After all, the more authors on a paper, the more likely it is that their total published research output would be larger, leading to higher outgoing references and consequently to higher self-citation counts, all other things being equal [34]. As before, the correlation between the number of coauthors and self-citations is indeed high ($r = 0.72$, $p < 10^{-4}$), until we omit the first two papers ($r = -0.15$, $p = 0.46$). Again, these two papers represent outliers both in terms of citations and self-citations, in contrast to prior findings that highly cited papers typically exhibit a lower rate of self-citations [33].

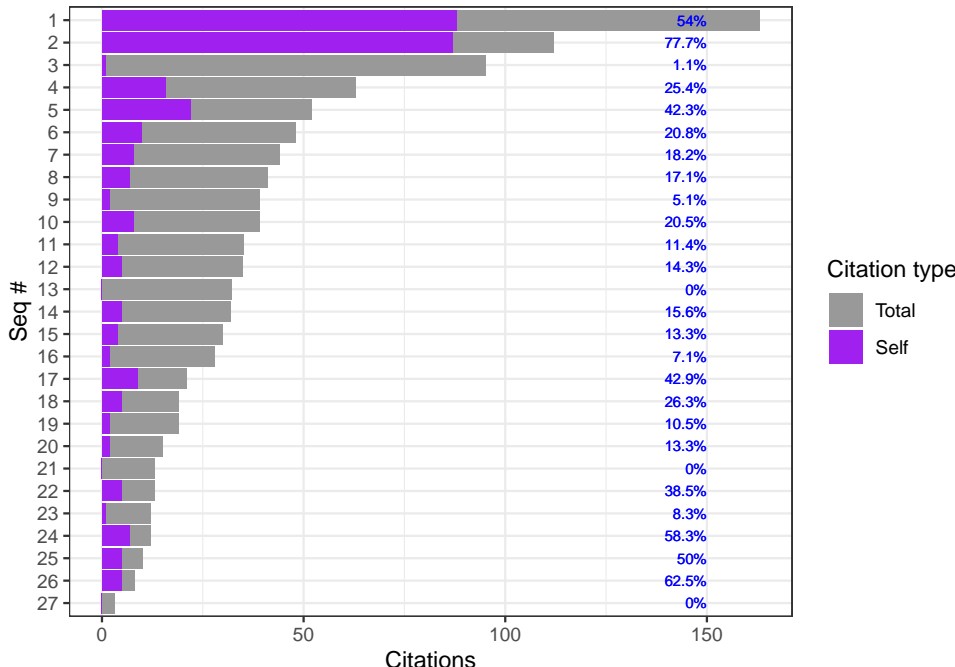

**Figure 5.** Self citations as a fraction of total citations after 5 years.

### 3.4. RQ4: What Keywords Are Associated with Higher Citations?

In this section, we look at the relationships between papers' citations and their text, specifically key terms. Extracting a meaningful list of words, or tokens, from each paper requires additional data preparation. First, the full-text must be converted from PDF format to text (ASCII), which involves first automated tools such as Linux's `pdf2txt` and then manual cleaning of poorly converted elements such as equations, tables, and formatting symbols. The references section, as well as conference and author details, are removed from the text since they contain many repeated and irrelevant keywords (such as "page"). Finally, the text is filtered to remove symbols, punctuation, and English stop words, and the remaining words are stemmed. This step was accomplished with the R package Quanteda [37].

To get a sense of the recurring terms in SIGMETRICS'17, Figure 6 shows the most common terms across all papers, with the size of each word weighted by its appearance frequency. The three most frequent terms are "time", "model" and "network", which appeared in all papers, except three papers missing "network". If we turn our attention instead to terms that are central to specific papers only, and not universally across the corpus, we can multiply the overall frequency of each term by its inverse-document-frequency to achieve the TF-IDF transformation [38]. As shown in Figure 7, the most focused term in the corpus is "DRAM", appearing 482 times in papers #1 and #2 and almost nowhere else, followed by "price", appearing 214 times across papers #4, #9, and #22. The contrast between these two metrics illuminates which keywords are more universal to SIGMETRICS'17 vs. central to specific papers.

Finally, to bring citations into the picture, we multiply the TF-IDF weight of each term by the log-transformed sum of citations of the papers containing each term:

$$\sum_{papers-with-term} log(citations(paper) + 1).$$

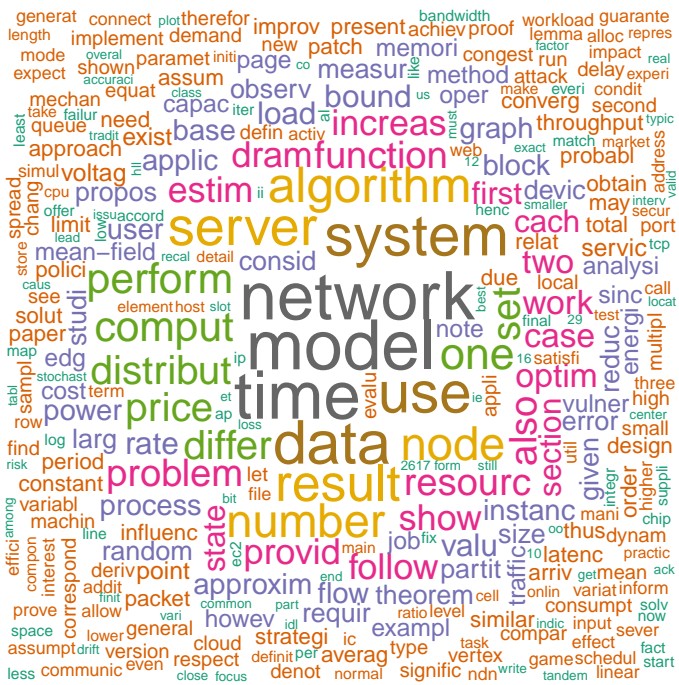

**Figure 6.** Word cloud of top terms by frequency.

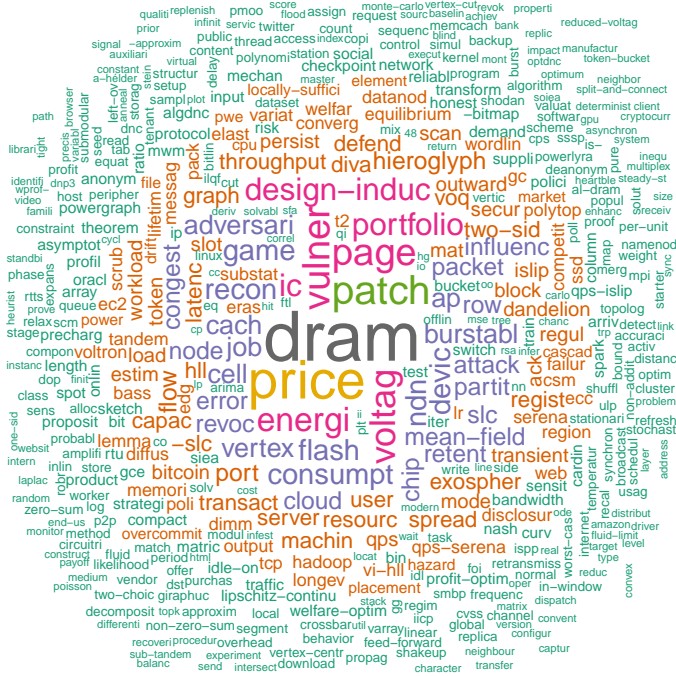

**Figure 7.** Word cloud of top terms by TF-IDF.

The logarithm function is used to attenuate the long-tail nature of five-year citations, transforming the distribution to a more linear weight [39]. As Figure 8 shows, there are fewer influential terms overall now, because there are few highly cited papers. The two most cited papers, #1 and #2, are also the ones that focus on DRAM, so not surprisingly, this term retains the highest weight. The term "price" is mostly split among three papers, some better-cited than others, which leads to a small reduction in its weighting. Overall, it appears from the figure that the most cited topics in SIGMETRICS'17 relate to memory,

energy consumption, and security. This set has nearly no intersection with the most common terms from Figure 6, such as "network", "model", "time", "data", "system", "server", and "algorithm", which are typical of performance evaluation, as we'd expect. The implication here is that while these latter terms may well characterize SIGMETRICS papers overall, they do not differentiate well between highly cited and moderately cited papers.

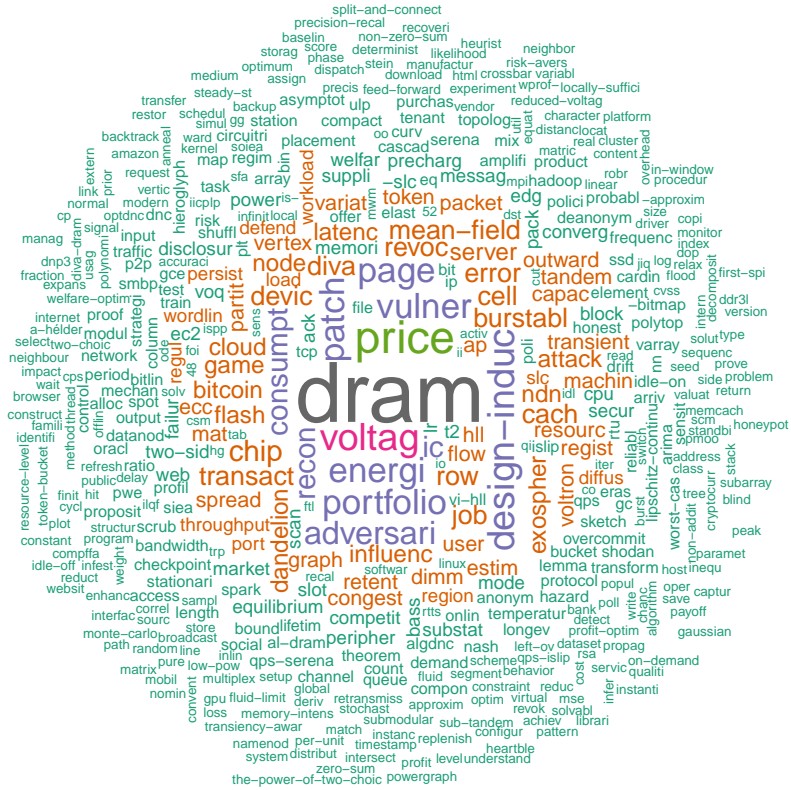

**Figure 8.** Word cloud of top terms by TF-IDF, weighted by log(citations + 1).

Other approaches to discerning key terms influencing citations include linear regression of citations by terms or classification of highly cited papers using Naive Bayes or decision trees. Unfortunately, none of these methods yielded useful insights on SIGMETRICS topics, fixating instead on nonspecific terms such as "region" and "relevance".

### 3.5. RQ5: What Other Factors Are Related to Citations?

We next turn our attention to various other metrics we can extract about the papers and compare them to citations. Some of these factors have been previously linked to higher citations, so we measured the Pearson correlation of each of these factors against the log-transformed citation count after five years.

For example, the open availability of papers can sometimes make a difference in citations, as alluded to in the introduction. Observing the duration it took GS to discover each paper or to publish a link to its e-print, there was very little variance between the papers in our dataset, which means we cannot validate this hypothesis for SIGMETRICS. However, there are multiple other hypotheses on factors that affect citations that we are able to test for our dataset.

#### 3.5.1. Textual Features

In a large meta-analysis from 2019 [40], Xie et al. observed a moderate correlation of $r = 0.31$ between paper length and citations. In our SIGMETRICS'17 dataset, the correlation between number of words in a paper and its log-transformed citations is somewhat higher

($r = 0.52$, $p < 0.01$). Our dataset is much smaller, which could partially explain the larger effect size, but it also controls for some influential external factors such as publication year, reviewer composition, and journal quality. We can speculate on causal arguments for this correlation, such as longer papers having more citeable statements or a greater diversity of data and ideas [41]. However, it is also quite possible that there are hidden confounding variables that themselves affect citations.

One such factor is the number of outgoing references per paper [41], which in our dataset is positively correlated with both paper length ($r = 0.52$, $p < 0.01$) and with citations ($r = 0.58$, $p < 0.01$). A 2009 study found a similar correlation of 0.44 in evolutionary psychology papers [39]. The same study also found a weak ($r = 0.2$) correlation between citations and the number of coauthors, as did Fox in 2016 [41], and we observed as well ($r = 0.38$, $p = 0.05$).

Another such factor is the readability of a paper, as measured by various statistics. For example, a 2019 study found that the Linsear–Write readability metric is negatively associated with the top quintile of cited Economics papers [42]. Recall that lower readability scores mean that the text is *more* readable, which implies that difficult-to-read articles may discourage citations. To measure the readability of SIGMETRICS papers, we again turn to the Quanteda R package, which offers at least 44 readability metrics. Of these, the metric that empirically correlated most strongly with log citations was "FOG.NRI" [43]. In our dataset, the overall correlation with citations is weak and positive ($r = 0.18$, $p = 0.36$); the least readable quintile does not show significantly fewer citations than the other 80% either ($t = 0.27$, $p = 0.8$). It is possible, however, that standard readability heuristics are not readily applicable to SIGMETRICS text because most papers include mathematical symbols, equations, figures, and tables that are not typically handled gracefully by such methods. For example, many of these heuristics assume that the shorter "words" between punctuation in mathematical formulae represent more readable text.

Even more esoteric textual features have been found to correlate with higher citations counts, such as the length of the title and its inclusion of a colon, either positively [30] or negatively [44]. In the SIGMETRICS dataset, neither title length ($r = 0.05$, $p = 0.8$), nor a colon in the title ($t = 0.27$, $p = 0.79$). are correlated with citations.

### 3.5.2. Peer-Review Features

We may also expect that papers that were rated highly enough in the peer-review process as to receive an award would fare well in citations, although rarely at the top [45–47]. SIGMETRICS'17 awarded a "Best Paper" award to paper no. 24 and a "Kenneth C. Sevcik Outstanding Student Paper" award to paper no. 20. As can be inferred from their relative rankings, these two papers fared well below the median SIGMETRICS'17 paper, with five-year citations at 12 and 15, respectively.

This peer-review perspective segues naturally to examining aspects of the review process itself. In an author survey we conducted in 2019 [48], authors from eight different SIGMETRICS'17 papers shared details about their reviews and the reviewing process. Although survey responses remain confidential, and the low number of responses is insufficient to draw statistically significant conclusions, we can still observe four trends in the responses. First, all eight papers received at least four reviews, and some as many as six, which is much high than the typical CS journal [49]; the mean number of reviews per paper at SIGMETRICS'17, 5.3, is even higher. Second, most respondents spent seven or more months on their projects prior to submitting to SIGMETRICS, and most chose SIGMETRICS as their first submission venue. Third, authors generally viewed the double-blind reviews favorably, labeling only two reviews of 23 as "unhelpful" and "missing major points", and labeling none as "unfair". Finally, perhaps the most salient anecdote is that there appears to be no correlation between the reviewers' mean overall paper grade and the paper's eventual citations. We speculate on possible reasons for this last point in the discussion section.

### 3.5.3. Demographic Features

Finally, we look at authors and examine the four demographic factors we collected on authors: gender, country, sector, and experience. The literature on the relationship between an author's gender and their eventual citations is mixed [50]. Of our 104 authors, 11 are women (10.6%). This low ratio is regrettably not unusual for systems conferences [51]. Comparing the median of citations across the two genders yields no significant differences ($W = 539.5$, $p = 0.77$), and neither does comparing the means of log-citations ($t = 0.33$, $p = 0.74$). We also found no evidence that men cite themselves more than women ($t = 0.05$, $p = 0.96$), unlike the findings in previous studies [50,52].

These nonsignificant differences repeat for geography when we compare log-citations of the nearly half of authors based in the US to the rest ($t = 0.47$, $p = 0.64$). There is also no apparent correlation between an author's citations and their research experience as expressed in past publications ($r = -0.27$, $p = 0.02$), past citations ($r = -0.17$, $p = 0.14$), or h-index ($r = -0.1$, $p = 0.39$). We could also compare paper citations only to the highest h-index per paper to control for variance within a research team, but even then, the correlation is nonexistent ($r = 0$, $p = 1$)

It is possible that the double-blind review process of SIGMETRICS—which ostensibly focuses on merit and limits the amount of variance in review outcomes based on author demographics—also limits the amount of variance in citation outcomes based on demographics.

The only significant difference in demographics-based citations in our dataset appears to be based on sector. The median citations for authors affiliated with industry (78) is more than double that of authors affiliated with academia (35) and the difference is statistically significant ($W = 608$, $p = 0.02$), as is the difference in mean log-citations ($t = 2.96$, $p < 0.01$). These statistics may not be as powerful as they sound, because after all, there are only 12 verified industry authors in this set. Nevertheless, it is worth noting that half of these industry authors appear on papers #1 and #2 and none appear in a paper ranked #15 or lower in total citations. Conceivably, the perspective or resources brought to performance-evaluation research by industry representatives receives an outsize proportion of the overall SIGMETRICS citations.

### 3.6. Regression Model

To summarize and aggregate our empirical findings, we can build a generalized linear regression model with log-citations as the response variable and various features we explored as predictor variables. The factors we included and their associated coefficients in the model are summarized in Table 3.

**Table 3.** Generalized linear model of log-citations as a function of paper factors.

| Feature | Coefficient | *p*-Value |
| --- | --- | --- |
| Intercept line | 2.2829 | 0.075 |
| Paper length (words) | 0.0001 | 0.220 |
| No. of references | 0.0160 | 0.275 |
| Title length (words) | −0.0390 | 0.580 |
| Colon in title | −0.5054 | 0.160 |
| Award winner | −0.4697 | 0.412 |
| Months to GS discovery | 0.0638 | 0.857 |
| Months to e-print | 0.0220 | 0.584 |
| Self-citations | 0.0085 | 0.611 |
| Maximum h-index of coauthors | −0.0095 | 0.279 |
| Any coauthor from industry | 0.1092 | 0.842 |
| Any woman coauthor | 0.3240 | 0.283 |
| Any coauthor from the US | 0.0477 | 0.888 |

This model has a McFadden $R^2$ value of 0.66, suggesting that it can explain about two thirds of the variance in citations. Nevertheless, none of the factors on their own exhibits a *p*-value below 0.1. Since some of these covariates are interdependent or collinear, we can try to improve the model and reduce the number of dependent factors using stepwise model selection [53]. The resulting model explains a little less of the variability in citations (McFadden $R^2$ = 0.59), but it captures it with fewer parameters: only paper length, no. of references, colon in the title, maximum h-index, and having a woman among the coauthors. Of these, the first two exhibit a p-value of under 0.05, and the third of 0.08. These three factors have been implicated by previous research findings as potentially linked with higher citations, which is encouraging to corroborate.

## 4. Discussion

Perhaps the most extraordinary finding about SIGMETRICS's citations is how ordinary they are, compared to similar contemporaneous conferences. The extremes on either end of the distribution are not so extreme and the variance is low, exhibiting a citation distribution that is more uniform than the other conferences'. None of the papers appear to have achieved the runaway exponential growth or slow decay in citations that are typical in so many fields, and most papers exhibit an atypical near-constant linear growth with no clear peak in the first five years. Additionally, aside from the top-two cited papers, most papers did not exhibit significant self-citations or focus on singularly highly cited terms.

In fact, if we treat the top-two cited papers as outliers and omit them, the citation picture appears more pedestrian still: the average citation count drops from 38.26 to 30.32, their growth drops from an average of 0.64 citations per month to 0.51, and the average fraction of self citations drops from 0.24 to 0.21. On the last point, it should be noted that four of the top-five cited papers in our dataset exhibit a significantly higher self-citations ratio compared to other papers. Without additional data on self citations from other SIGMETRICS years or other conferences, it remains unclear how characteristic this phenomenon is, a question we plan to investigate in a future study.

Most other perspectives we examined, including author demographics and paper readability, surfaced mostly negative findings, that is, a lack of relationship between these factors and the paper's five-year citations. Notable exceptions were some linguistic features, like the length of a paper or of its reference list, and the number of coauthors. These factors have all been associated with higher citations in past studies of other fields, so may not necessarily suggest that SIGMETRICS is unique in this way. The one positive association that may be unique to SIGMETRICS, at least in its magnitude, is the beneficial contribution of industry authors to a paper's eventual citations.

Another interesting anecdotal observation in our dataset is that citations show no strong association with review scores (and related, with paper awards, which in turn imply high review scores). This ostensible independence between the two is surprising not only because it contradicts previous research findings, but also because we might expected well-reviewed papers to exhibit the originality and interest that eventually translates to higher citations. The intuition behind this expectation is that both review scores and citations are quantities that try to approximate the same ephemeral, impalpable quality, the "goodness" of a paper.

Several hypothetical explanations to this discrepancy come to mind, including the small sample size, the notorious difficulty in trying to evaluate the "goodness" or "cite-ability" of a paper, the inherent noise in the review process [54], or that the two metrics measure different qualities after all. It is also possible that SIGMETRICS program committees explicitly value qualities other than perceived or predicted citeability. Investigating these hypotheses is another interesting venue for future research, but it may require a much larger dataset that includes information on rejected paper, which is not readily available.

*Threats to Validity*

Because of the time it takes to stabilize citation statistics, we opted not to include additional data from more recent years in our dataset. Undoubtedly, more data could strengthen the statistical validity of our observations; but it could also weaken any conclusions based on the inherent delays of the citation process and in variation over time. Moreover, our methodology is constrained by the manual collection of data. The effort involved in compiling all the necessary data and additional factors limits the scalability of our approach to additional conferences or years. Furthermore, the manual assignment of genders is particularly prone to human error. Nevertheless, such errors appear to be smaller in quantity and bias than those of automated approaches, as verified in our prior work on gender [51].

## 5. Conclusions and Future Work

We set out to explore five research questions on the citation behavior of SIGMETRICS's papers, focusing on a case study from 2017. These questions cover disparate perspectives, including textual features, author demographics, peer review, self-citations, and evolution over time. In the order of the questions, our main findings were:

- *All* SIGMETRICS'17 papers collected some citations from external sources, compared to other related conferences and more generally, scientific papers, where a sizeable proportion of papers remains uncited.
- On the flip side, *none* of the SIGMETRICS'17 papers achieved runaway success in terms of citations, especially compared to other contemporaneous conferences on the same topic.
- Most papers exhibited a near-constant citation velocity, again defying the common expectation of an accelerating increase in citations followed by a gradual decline.
- With the exception of the two most-cited papers, self-citations do not appear to be a significant source of citations for SIGMETRICS'17 papers.
- There appears to be no particular "buzzwords" among this set of papers that are associated with a particularly high citation rate.
- Among multiple factors examined in a linear regression model, none were found to be significant predictors of higher citation on their own. With all factors combined, however, the model predicts approximately two thirds of the variance in citations.

This work can be extended in many directions. As mentioned in the previous section, The interesting question of self-citations across fields and conferences remains open, as does the question of the ostensible discrepancy between review scores and citations. Additionally, we could look deeper into demographic factors of authors, such as their career stage or self-identified gender (and more generally, authors' intersectional identities). We could also compare conferences to journals, which typically exhibit slower initial exposure and citation, and are less commonly preferred in computer science.

Another extension of this work, one that should prove much easier, would be to observe the same dataset over longer periods of time, in an attempt to identify citation peaks and eventual changes in citation rates. We plan to follow up by updating the data repository with fresh citation data at regular intervals, and once we identify enough significant changes (or their absence), analyze and report these as well.

**Funding:** This research received no funding.

**Data Availability Statement:** All of the code and data for this article are publicly available at https://github.com/eitanf/sysconf [55].

**Conflicts of Interest:** The author declares no conflict of interest.

## Abbreviations

The following abbreviations are used in this manuscript:

| | |
|---|---|
| ACM | Association for Computing Machinery |
| CS | Computer Science |
| GS | Google Scholar |
| ICPE | International Conference on Performance Engineering |
| IMC | Internet Measurement Conference |
| SIG | Special-Interest Group |
| SIGCOMM | ACM's Special-Interest Group on Data Communication |
| SIGIR | ACM's Special-Interest Group on Information Retrieval |
| SIGMETRICS | ACM's Special-Interest Group on Performance Evaluation |
| SIGMOD | ACM's Special-Interest Group on Management of Data |
| TF-IDF | Term Frequency–Inverse Document Frequency |

## Note

[1] Since not all conferences have had their five-year anniversary yet, we use 54-month citations as the baseline for comparison.

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
