# Peer review of "Multifactor Citation Analysis over Five Years: A Case Study of SIGMETRICS Papers"

_publications, doi:10.3390/publications10040047_

Round 1

Reviewer 1 Report

The article suggests a current and attractive topic for academia. The effort is evident but requires adjustments, both in the results and methodology. I hope you will find the following observations useful: 

1) The footnotes (1 -2) can be inserted as references using web page type.

2) You should state the general aim you present in your study. Use the word "aim".

3) It is possible to summarise what is indicated in materials and methods with a figure... However, it would be fascinating to see the methodology summarised.

4) The limitations section should be at the end of the manuscript.

5) The Ethics statement section (lines 111-115 ) was indicated in lines 426 - 428. Please delete the indicated section. 

6) Lines 128 - 131 mention a procedure not reported in the statics section.

7) Table 1 should include a reference column so readers can locate the papers (is this possible?).

8) Table 2 should be separated from figure 2. The figure needs to be enlarged. For the conferences mentioned, it is necessary to indicate their full names, as some readers may not be familiar with them.

9) Figures 3 - 5 need to be enlarged. It is difficult to read them with these dimensions.

10) The software and procedures indicated in 3.4 and 3.5 are not in the methodology section.

11) The following sentence "network" (appearing in all but three papers)" is not understood.

12) Line 270 Xie et al. should go with the respective reference, not at the end of the sentence.

13) Sections 3.5 and 3.6 mentions the regression model but do not show it.

14) Discussion and conclusions should be separate.

15) Lines 421-422 are wrong; use the journal rules.

Author Response

Thank you for your valuable feedback. Responding to your comments in order:

  1. I've converted the footnotes to citations as requested.
  2. The following sentence has been added to the introduction: "...the aim of this study is to understand the specific citation patterns of this distinctive conference, and compare it to similar conferences."
  3. I'm afraid I don't understand this suggestion. It says: " It is possible to summarise what is indicated in materials and methods with a figure... However, it would be fascinating to see the methodology summarised."   I don't understand how the methodology can be summarized with a figure or how to summarize the methodology further in this rather short section.  Nevertheless, I've added the following summary sentence to the beginning of the section: "To answer these research questions, we collected citation data at regular intervals from a set of 2017 conferences, on which we performed various statistical analyses." If you have additional concrete suggestions, kindly share them with me.
  4. The Limitations subsection has been moved to the Discussion section as requested.
  5. The Ethics Statement has been removed as requested.
  6. I've added the linear regression method to the Statistics subsection as requested.
  7. I would rather not cite all the SIGMETRICS'17 papers in Table. 1. Not only will this inflate the reference list of this paper and the papers' citations gratuitously (since this paper doesn't extend their research), but it will also distort the citation picture for exactly the kind of study that this study is, trying to evaluate how many "real" citations they get.
  8. In my experience with this journal, the final sizing of the figures and the proximity of Fig. 1 to Table 2 will all be formatted by the journal staff. There is no need to perfect the draft submission as the final formatting will look very different anyway. I did, however, add the expansion of the acronyms in Table 2 to the abbreviations list as you suggested.
  9. See the previous comment on journal's formatting. Nevertheless, I've enlarged all figures.
  10. I've added a description of the programming environment to the methodology section, and where to find specific details (the available source code).
  11. I've reworded the sentence in the hope of making it easier to understand.
  12. I've moved the citation closer to the reference, as requested.
  13. The regression model is actually completely described in Table 3. Kindly clarify what else needs to be shown.
  14. I've separated the conclusion and future work from the discussion and edited it a little.

Reviewer 2 Report

Thank you for the opportunity to read your manuscript, which I very much enjoyed. I feel overall this is a strong piece with an interesting perspective and insights to offer to readers. I would concur with your statement of originality, and focus. Additionally, I found the prose and author voice throughout to be an engaging one, which encouraged me to read on while following the direction of your arguments. The data analysis certainly presents robust and detailed examination of the subject matter too.

Further observations

Lines 90-100: I was interested in your comments on binary gender perceptions, and wondered if there was any value to considering any work looking at gender identities beyond these normative male/female ones. However, I suspect this is likely beyond the remit of the paper. Perhaps fodder for future work in this domain?

Lines 116-126: I appreciated the observations and rationalisations of the limitations within the study.

Lines 371-391: I was intrigued by this finding, and perhaps it is the cynic in me, but your surprise at the limited ‘success’ of the works in terms of citations is itself surprising. My personal experience of works has been it is a rare paper which escalates to the heights, and that the vast majority of work receives such ‘pedestrian’ citation acceleration and totality. Perhaps it may be of, some, value in seeking out a recent paper or two that considers what the ‘normative’ experience is for citations within this (or ancillary) fields; although I suspect such a reductive or averaged experience may prove elusive.

Author Response

Thank you for your valuable feedback! To answer your points in order:

  • Looking into self-identified gender (and more generally, intersectional demographics) could indeed prove highly insightful. Unfortunately, as you suspect, such information is not generally available for published authors, and even conferences who survey their authors to collect demographic information rarely share more than cursory statistics about the distributions. For a more meaningful treatment of this topic, please revisit our papers "Gender Differences in Collaboration Patterns in Computer Science" in this very journal and the PLOS One paper (reference [24] in this paper).
  • The "surprise" expressed about SIGMETRICS is that its citations curves are relatively flat, both compared to the other conference data collected and shown in the paper, and compared to the (somewhat meager) information about related conferences in other papers (references [2--5]). The long-tailed distribution of citations you describe is indeed the common case. What's surprising about SIGMETRICS is how relatively unskewed the distribution is, compared to the other conferences. The paper's text has been augmented to sharpen this point.

Round 2

Reviewer 1 Report

This interesting manuscript suggests a current and attractive topic for the academy. The effort made by presenting an interesting document that has undergone some modifications due to the evaluators' recommendations has allowed us to have a novel and rich document on a complex subject.

I appreciate the author's patience when considering the vast majority of the recommendations made with quality and professionalism. The methodology and the data set as its analysis are solid. The conclusions are relevant.

My sincere congratulations to the author for this important contribution to academia. I consider the article to be publishable.

Author Response

Thank you for your kind words!